# Parental involvement in school-based mental health interventions for young people in low-resource settings: A qualitative study from Zimbabwe and Ghana

Rufaro Hamish Mushonga[1,2]*, Rebecca Jopling[1], Franklin Glozah[3], Tiny Tinashe Kamvura[4], Suzanne Dodd[1], Denford Gudyanga[1], Arnold Maramba[4], Edith Dambayi[5], Christopher Abio Ayuure[6], Tarisai Bere[7], Fabian Sebastian Achana[5], Lucy Owusu[1], Dixon Chibanda[4], Melanie Abas[1], Benedict Weobong[3], Moses Kumwenda[2]

**1** Institute of Psychiatry, Psychology and Neuroscience, King's College London, England, **2** Department of Pathology, Kamuzu University of Health Sciences, Blantyre, Malawi, **3** School of Public Health, University of Ghana, Accra, Ghana, **4** The Friendship Bench, Harare, Zimbabwe, **5** Navrongo Health Research Centre, Ghana, **6** Ghana Health Service, Ghana, **7** Department of Mental Health, Faculty of Medicine and Health Sciences, University of Zimbabwe, Harare, Zimbabwe

* rufaro.mushonga@kcl.ac.uk

## Abstract

### Background

Young people in low-resource settings are disproportionately affected by mental health problems, yet access to formal mental healthcare remains limited. However, schools in these regions have increasingly become the primary settings for mental health interventions, offering an accessible and supportive environment for mental health services. Recognising the critical role schools play in providing mental health services for young people, there has been a growing emphasis on involving parents in school-based mental health (SBMH) interventions in low-resource settings. This study explored the mechanisms for effectively involving parents in SBMH interventions for young people in Zimbabwe and Ghana.

### Methods

Cross-sectional qualitative research was conducted in Harare, Zimbabwe and Navrongo, Ghana. This study is a sub-analysis of a larger formative qualitative study which aimed to identify evidence for adapting interventions for depression and anxiety for young people aged 15–24 in Zimbabwe, and 15–18 in Ghana, and test the feasibility of the adapted intervention. We utilised semi-structured in-depth interviews, key informant interviews and focus group discussions with various stakeholders until data saturation was achieved. All interviews were audiotaped and later transcribed verbatim and translated to English for analysis. Data for this study were inductively coded and analysed using thematic analysis.

**Data availability statement:** In accordance with the ethical approvals, access to the data is restricted to the study team, who are co-authors of this paper. As participant consent was not obtained for data sharing beyond the study team, the authors are unable to deposit the data in a public repository or share it with individuals outside the project. The data underlying the results presented in this study are available from the Institute of Psychiatry, Psychology & Neuroscience at King's College London. Data access requests should be direct-ed to research.data@kcl.ac.uk.

**Funding:** This work was funded by the National Institute for Health Research (NIHR) through 'African Youth in Mind', a Global Health Research Group, led by King's College London.

**Competing interests:** The authors have declared that no competing interests exist.

## Results

Effective mechanisms for engaging parents in SBMH include routine parent-teacher meetings, interface meetings between parents and school-based mental healthcare providers, and direct parental participation in sessions. However, while parental involvement is key for SBMH interventions, it can be problematic. In some cases, parents may unintentionally breach their children's privacy and confidentiality or may be the source of their children's mental health problems.

## Conclusion

The study's findings underscore the importance of parents as vital partners in SBMH interventions. Given the positive impact of parental involvement, it is essential to incorporate parents into the design and implementation of these interventions. By leveraging the insights from this study, interventionists can develop and implement more effective and low-cost SBMH interventions, which can significantly improve mental health outcomes for young people in low-resource settings.

## Introduction

Common Mental Disorders (CMDs) like depression and anxiety are among the leading causes of impaired health and diminished daily functioning globally [1,2]. Prevalence of common mental disorders in young people under the age of 24 is high, with 20% suffering from common mental illnesses globally [3]. In sub-Saharan Africa (SSA), young people between ages of 15–24 constitute approximately 13.3% of the total population [4] and are particularly at higher risk of CMDs [5]. A recent systematic review estimates the prevalence of mental health issues among this demographic to be 27% for depression, 30% for anxiety disorders, and 12% for suicidal ideation [6].

While significant progress has been made in the provision of treatment for CMDs in SSA-including task-sharing like the Friendship Bench intervention [7], introduction of transdiagnostic approaches [8], and the decentralization of mental health care, challenges persist [9]. In this region, mental healthcare is still inadequate, ineffi-cient, and inequitable [10], leaving a 90% treatment gap for CMDs [11,12]. This is mainly attributed to lack of funding and unavailability of trained specialists to provide mental health services [8]. Consequently, most young people in SSA are left with no option, but to live with their undiagnosed and untreated mental disorders or seek help from alternative sources such as religious or traditional healers. In Zimbabwe, research conducted by the United Nations Children's Fund (UNICEF) indicates that at least 27% of young people aged 15–24 suffer from depression [13]. However, the true burden of mental health issues is likely higher among young people in Zimba-bwe due to compounding factors like the COVID-19 pandemic, rising poverty, high unemployment rates and the economic crisis [14]. In Ghana, a 2023 study on mental health in primary care settings found the prevalence of depression among young people to be 59.2% [15].

In response to the growing mental health needs of young people, especially in low-resource settings like Zimbabwe and Ghana, schools have increasingly become critical providers of mental health services [16–18]. Evidence shows that schools offer a structured and supportive environment where large numbers of children can be reached, as they help reduce barriers to access such as transportation and financial constraints, especially in low-income settings [19,20]. Recognising the key role schools play, Zimbabwe's Ministry of Health and Childcare introduced the "Catch Them Young" program. As part of the 2020 WHO-Special Initiative for Mental Health, this program is a pilot of mental health screening and psychological first aid support in schools [21]. It aimed to empower teachers to support pupils during and after the COVID-19 pandemic. In addition, the Zimbabwe School Health Policy (ZSHP) provides a framework for addressing various health-related issues concerning the welfare of learners in the school system, including mental health, as well as provisions for care, support, guidance, and counselling needs of all learners [22]. In contrast, Ghana, with its provision of free senior high school education, implements a pioneering national adolescent health policy. School nurses visit approximately three days a week to provide clinical care to pupils and refer to nearby health facilities where necessary. However, while mental health is listed as one of the Ghana school health packages [23], it remains least developed.

In addition to School Based Mental Health (SBMH) interventions for young people in low-resource settings, there has been a growing emphasis to involve parents in these programs [24–26]. Parental involvement, which entails active engagement of parents with their child [8], significantly influences their physical, mental, and social development [27]. Research indicates the positive impact of parental involvement in SBMH interventions for young people, including improved mental health outcomes [20,24,25,28,29]. However, most of this research comes from high-income countries, creating a gap in low-and-middle-income-countries (LMICS) like Zimbabwe and Ghana, where such interventions are urgently needed. Notably, parental involvement is currently not a formal component of SBMH interventions in either Zimbabwe or Ghana, highlighting a crucial gap in existing programs. To bridge this gap, the present study explored the views of various key stakeholders to understand how parents could be involved in SBMH interventions in these two countries. The insights gained from our study will contribute to developing appropriate and effective strategies of engaging parents in interventions aimed at improving youth mental health outcomes in low-resource settings.

## Materials and methods

### Study context and design

A cross-sectional qualitative research design was adopted for this study. The study was conducted as part of formative work for the 'African Youth in Mind' consortium, which aims to identify evidence to adapt interventions for depression and anxiety for youth aged 15–24 in Zimbabwe and 15–18 in Ghana, and to test the feasibility of the adapted intervention. The difference in age groups between the two countries reflects the distinct focus of each setting. In Ghana, schools were specifically chosen as study sites, aligning with the school-going age range and the project's efforts to strengthen existing school-based mental health systems. In contrast, in Zimbabwe, the study was centred on clinics to explore how to adapt the Friendship Bench model for youth mental health support. In reporting this study, we adhered to the COREQ (COnsolidated criteria for REporting Qualitative research) guidelines [30].

### Study setting

In Zimbabwe and Ghana, the study was conducted in Harare and Navrongo, respectively. Harare, Zimbabwe's capital has a population of about 1.6 million. Like elsewhere across the country, poverty and unemployment are the main contributors to mental health issues in Harare, particularly among young people [31]. Data collection in Harare took place between 20 December 2022 and 30 September 2023. Navrongo is a town in the Upper East Region of northern Ghana with a population of about 100, 000, and has a high prevalence of CMDs among young people. This is largely due to widespread poverty, limited access to mental health services, seasonal food insecurity, high youth unemployment, and restricted educational and economic opportunities, all of which can negatively impact young people's mental well-being [32]. Data

collection in Navrongo took place between 1 January 2023 and 31 May 2023 in the Navrongo Health and Demographic Surveillance Site (NHDSS) area.

## Study team

In Zimbabwe, the interviews were conducted by three research assistants (RAs) (two males and one female) who had a minimum bachelor's degree in social sciences. The RAs, bilingual in Shona and English, had vast prior experience in collecting qualitative data. In Ghana, the interviews were conducted by four research assistants. The RAs (one female and three males all multilingual in Kasem, Nankani and English) had a minimum of a bachelor's degree with two to five years' experience in conducting qualitative research. Prior to data collection, all the RA's received initial training on study aims and objectives, study protocol, data management, participant recruitment, qualitative research skills and research ethics.

## Sampling and recruitment

In both countries, purposive and snow-ball sampling techniques were used to recruit study participants. Invitations were extended using information sheet and informed consent procedures, with assent obtained for those under 18. In Zimbabwe, key policymakers, including the Director of Mental Health Services at the Ministry of Health and Director of School Psychological Services at the Ministry of Education were initially approached and asked to recommend other key informants. This resulted in the recruitment of 5 policymakers. In Ghana, the Municipal and District Directors of Health and Education Services at the Ghana Education and Health Ministries were similarly approached, leading to the recruitment of 5 key informants each (3 in the Kasena-Nankana Municipality and 2 in the Kassena-Nankana West District) respectively. All schools in Zimbabwe were public institutions located in high-density suburbs of Harare, while in Ghana, recruitment was conducted in rural schools. Thus, both settings are characterised by limited resources and financial constraints. School staff in both countries were recruited through recommendations from school heads, totalling 39 staff members. School-going young people were recruited with permission from school heads, resulting in the recruitment of 43 students from both countries. Healthcare workers were recruited by approaching nurses in charge at district hospitals, who recommended other informants resulting in 22 recruited. Lay health workers in Zimbabwe were identified through their supervisor who recommended other delivery agents involved in the Friendship Bench intervention, resulting in 20 recruited. In Ghana, similar methods were used, resulting in 6 recruited through health care centres. Caregivers of young people receiving mental health services were invited to participate, leading to 14 recruited. Young people with lived experience of poor mental health were identified through their involvement with mental health services, with 15 recruited in Zimbabwe through the Friendship Bench, and 12 in Ghana through health care centres.. Finally, faith-based and traditional healers were also recruited through mental health coordinators who were well acquainted with them, resulting in 6 recruited. The information is summarised in Table 1 below.

## Inclusion and exclusion criteria

Inclusion criteria for young people were 15–24 years in Zimbabwe and 15–18 years in Ghana, and able to provide informed consent or assent for those under 18. Those with lived experience had received formal care for symptoms of depression and have not responded to problem-solving therapy (PST) as defined by less than a 2-point reduction on the SSQ14 from their baseline score. Professionals or volunteers (health care workers, faith-based and traditional healers, policy makers, school staff) aged 18 and above working in various capacities to deliver or implement mental health services, and those willing and able to take part in the study were included. Exclusion criteria included individuals working to deliver or implement mental health services for young people but had not been working or living in the community for at least 12 months. Moreover, in both countries, young people with lived experience of depression were excluded if they had an active major mental health condition, determined by a previous clinical diagnosis of a severe mental health condition that could impair their ability to engage in the study. Additionally, individuals who were actively suicidal (assessed through

**Table 1. Socio-demographic profile of participants.**

| Category | Zimbabwe | | | Ghana | | | Total |
|---|---|---|---|---|---|---|---|
| | N | Sex | | N | Sex | | |
| | | F | M | | F | M | |
| Care Givers | 10 | 9 | 1 | 4 | 3 | 1 | **14** |
| Health Care Workers | 4 | 4 | 0 | 18 | 10 | 8 | **22** |
| Lay Health Workers | 20 | 20 | 0 | 0 | 0 | 0 | **20** |
| Policy Makers | 5 | 3 | 2 | 10 | 3 | 7 | **15** |
| Young People who received mental health services | 15 | 7 | 8 | 12 | 8 | 4 | **27** |
| Young People who did not receive mental health services | 10 | 5 | 5 | 33 | 17 | 16 | **43** |
| Faith-based healers | 0 | 0 | 0 | 2 | 2 | 0 | **2** |
| Traditional healers | 0 | 0 | 0 | 4 | 1 | 3 | **4** |
| School Staff | 13 | 9 | 4 | 26 | 6 | 20 | **39** |
| Total | **77** | **57** | **20** | **109** | **50** | **59** | **186** |

P4 screener) or had an advanced physical illness or impairment (assessed at informed consent procedures) that would interfere with their ability to participate were excluded.

## Procedure

Key informant interviews (KII) were conducted to collect data from policy makers, school staff, healthcare workers, caregivers, faith-based and traditional healers and lay health care workers. For young people, in-depth semi-structured interviews were conducted. Additionally, in Ghana, separate focused group discussions (FGD) with students who had no mental health history were conducted. The interview guides included a core list of open-ended questions and anticipated follow-up questions. This ensured that the RAs asked all participants a minimum set of identical questions to collect credible and comparable data. In Zimbabwe, interviews were conducted in English and Shona, and in Ghana interviews were conducted in English, Kasem and Nankani. Data collection continued until data saturation, where no new information was emerging [33]. All interviews, lasting an average of 45–60 minutes were conducted face-to-face, and were audiotaped and later transcribed verbatim. Transcriptions were translated from Shona (Zimbabwe) and Nankani/Kasem (Ghana) to English for analysis. Transcripts underwent quality checks and were pseudo-anonymised to ensure the confidentiality of participants' identities.

## Data analysis

Data for this study were inductively coded using thematic analysis, a method designed for identifying, analysing, and reporting patterns (themes) within the data [34]. Consortium team members, other than the transcribers conducted checks on the transcripts to ensure quality and accuracy. An initial codebook was generated from the topic guides and first interview transcripts. Nvivo software (Version 14) was employed to code and organize the data. Each transcript was assigned to at least two coders, who read the respective transcript to familiarise themselves with the data. Coders added new codes where necessary. These new codes were merged and regular intervals using the Nvivo 14 collaboration cloud function, to ensure all coders were able to use the same codebook. Regular study team meetings (bi-weekly) were held to discuss progress in coding as well as the evolving codes and themes. Codes were reviewed and discussed with an additional round of coding conducted to ensure codes were specific and consistent before patterns in codes were explored to develop themes. A group of at least 3 researchers, including one senior researcher (MK and FG) worked together to review codes and finalize the themes.

## Ethical considerations

The study was approved by the respective institutional ethics committees in Ghana, Zimbabwe and the UK (at the setting of the lead organisation) and all participants provided written informed consent prior to their participation. For those under 18, written informed consent was obtained from their parents or guardians, with assent also obtained from minors. The study was approved by the Medical Research Council of Zimbabwe (Ref: MRCZ/A/2965), Navrongo Health Research Centre Institutional Review Board (Ethics Approval ID: NHRCIRB480), Ghana Health Service Ethics Review Committee (GHS-ERC:026/07/22) and the Kings College London Research Ethics (Ref: HR/DP-21/22–32917). To gain access to selected schools, approval was also sought from the Ministry of Primary and Secondary Education in Zimbabwe, and the Ghana Education Service in Ghana.

## Inclusivity in global research

Additional information regarding the ethical, cultural, and scientific considerations specific to inclusivity in global research is included in the Supporting Information (S1 Checklist).

## Results

Four distinct themes emerged from qualitative analysis of involving parents in SBMH interventions for young people. These themes included 1) routine meetings between teachers and parents, 2) interface meetings between parents and school-based mental healthcare providers, 3) direct participation of parents in sessions, and 4) negative aspects of parental involvement.

### Routine meetings between teachers and parents

Participants emphasised the importance of routine meetings between teachers and parents as an essential platform for the dissemination of information about mental health and the interventions available. These parent-teacher association (PTA) meetings, as participants noted, provide an opportunity for a broad spectrum of parents and teachers to collaborate on addressing mental health concerns, raise awareness about the intervention as well as building trust and open communication. One participant noted:

*The only way you can get most of the parents is usually when there is a PTA meeting. So that is a way that you can get them and give them that education concerning mental health issues. In some of the PTA meetings, sometimes school heads invite professionals from the mental health unit to draw the attention of the parents to know such issues and how to help them: Senior High School Teacher, Ghana.*

In addition, another participant noted

*I think you can invite them to school to talk about some of the life issues that the children are going through. Because sometimes they [young people] would understand it better when they are told by their own parents and that could be effective: High School Teacher, Zimbabwe.*

Young people also shared the same sentiments, highlighting integrating mental health awareness in routine meetings between teachers and parents as key for raising awareness about mental health challenges they may be facing.

*Our school has PTA meetings, so if they could bring parents to raise awareness about mental health issues we might be facing: Senior High School Student, Ghana.*

From these perspectives, routine PTA meetings serve as a vital bridge between parents and teachers, facilitating the dissemination of information and fostering collaboration necessary to address young people's mental health issues. By involving both parents and professionals in these discussions, schools can enhance awareness, support, and intervention strategies, ensuring that young people receive the comprehensive mental health care they need both in and out of the classroom. While PTA meetings were a well-established and widely utilised structure for parental engagement in Ghana, Zimbabwe had less formalised PTA structures. This difference may influence the feasibility of using PTA meetings as a primary channel for engaging parents in SBMH initiatives. As a result, alternative strategies may be required in contexts where PTA participation is limited.

## Interface meetings between parents and school-based mental health providers

Study participants also mentioned that involving parents in school based mental health intervention can be effectively achieved through interface meetings between parents and school-based mental healthcare providers. Unlike PTA meetings which involve general discussions between parents and teachers, interface meetings between parents and school-based mental health providers focus on direct collaboration between mental health professionals and parents. These meetings are more specialised, aimed at addressing specific individual mental health concerns and creating tailored solutions. One policy maker stated:

*I think they [parents] should be invited to a meeting because they need to understand the whole concept [the intervention] then they know where they are going and what is the likely impact on their children. They need information about the whole package for them to be very useful in the intervention:* Policy maker, Ministry of Education, Ghana.

This shows that such meetings are essential for parents to grasp the scope and purpose of the intervention, thus enabling them to support their children more effectively. Also, young people felt that interface meetings are key for parents to understand their children's mental health issues, which can prevent unnecessary blame and conflict, as one participant remarked:

*You can call them for a meeting and talk to them about some of these things. Like explaining to them that, for example, if you see your daughter sitting down lonely and sad, you can approach her and know how to handle them. They should not always be blaming their children unnecessarily, sometimes their children are also right:* Senior High School Student, Ghana.

In addition, another participant with a mental health history expressed that parental involvement can help to sensitise the parents on the mental health issues that they might be facing:

*Parents wouldn't know what their children are going through. They are not even aware. They will just say its stress. They just take it for granted. So they should attend those meetings:* Young person with a mental health history, Zimbabwe.

From these excerpts, interface meetings serve as vital platforms for direct collaboration, personalized support and guidance tailored to specific needs of young people with mental health issues. Yet, these meetings not only empower parents to be more understanding and supportive, but also help to alleviate potential sources of mental health concerns within the home setting.

## Direct participation of parents in the sessions

Finally, direct participation of parents in therapy sessions was emphasised by participants for several reasons. Firstly, it ensures that parents are aware of their child's progress and can contribute valuable insights or concerns about their

child's mental well-being. This collaborative approach ensures that underlying causes are addressed effectively. One caregiver expressed:

*As a parent you would like to hear what your child is saying during the discussion between the child and the caregiver, so that you can tell if they are really understanding each other and if s/he is really speaking from the heart:* Caregiver, Zimbabwe

Moreover, their presence in sessions is key to gauge the effectiveness of the intervention and ensure that their child's mental health needs are being adequately addressed. This was echoed by a school-based mental health provider:

*When I am dealing one on one with a child, I want their parents to be present. So, that if there are issues the parents will want to bring, the parents will be able to help:* Senior High School Nurse, Ghana.

In addition, parents can also reveal information that their child may be reluctant to share or be unaware of, thus 'filling in the gaps'. This was revealed by one participant with a mental health history:

*Maybe the student might be hiding something that the parents know she/he doesn't want to voice out and that particular thing might also be part of the solution. So, I think if you call the parent, the parent can also tell what the student is hiding:* High school student with a mental health history, Ghana.

Another participant echoed the same sentiments

*The parents must be involved, and the parents too must explain to the counsellor or the person in charge details about the depressed person for the counsellor to know exactly what caused the depression:* Nurse, Zimbabwe.

However, there are different levels of parental involvement suggested by participants, i.e., the necessity of involving parents in sessions varies with the severity and nature of the student's condition. For minor issues, parental involvement might not be necessary, but for more serious cases, informing and engaging parents becomes important. This could be just informing the parents if there is an issue with their child and informing them the school's observations and planned intervention. Two participants expressed:

*There are some mental health conditions that you don't need to really involve the parents because of the distance to the school:* Senior High School Teacher, Ghana.

*It depends on the level of the case or the acuteness of the case. If it is just a minor condition, you don't need to involve the parents but if it is a serious one and it is damaging to the person, you can involve them:* Mental Health Nurse, Ghana.

These suggestions ensure that the level of parental involvement is appropriately tailored to the severity of the mental health issues, while practical factors such the distance from parents' homes to school are considered.

### Negative aspects of parental involvement

The involvement of parents in the SBMH interventions for young people, while beneficial as highlighted above, can also be problematic. In some cases, their involvement may compromise young people's privacy and confidentiality, or they may be the source of mental health problems, thus making it hard for young people to open up.

**Privacy and confidentiality concerns.** Some participants in the study expressed concerns about involving parents in SBMH interventions, fearing that it would compromise their privacy and confidentiality. One caregiver of a young person with mental health history expressed:

*Children may want privacy, and when they realize that their parents are getting to know what they are into, some may choose to be alone:* Caregiver of a young person with mental health history, Ghana.

This quote suggests that parental involvement could deter young people from seeking help, especially when they realise that their privacy is at stake. Additionally, another caregiver emphasised the importance of considering young people's autonomy.

*They [parents] mustn't be given all the information. Teenagers are now able to stand on their own. However, a parent is needed sometimes for those who are disabled and those on ART, those must continue dealing with parents: Caregiver, Zimbabwe.*

This sentiment underscores the view that young people should have autonomy, and exercise their agency in deciding which information should be shared with their parents. However, those with chronic conditions such as those disabled or on antiretroviral therapy (ART) may require continued parental involvement. This shows that autonomy is not absolute, but rather contextual, where young people exercise autonomy in certain areas but still rely on parents or caregivers for more complex situations.

Others participants mentioned that parental involvement can lead to discomfort particularly when parents begin to be too intrusive.

*I don't think that will work out; they should not be involved. Because it will be difficult, they start asking questions, like why are you doing this, and you yourself might not be comfortable saying what you really want to say:* Senior High School Student, Ghana.

As a result, some young people prefer confiding in non-parental figures, such as peers and counsellors, due to lack of a comfortable relationship with their parents.

*When someone is not my peer, I can confide in them and then get comfortable because they don't know me. You can't say something in front of my parents. Yeah, not all of us have that kind of relationship with our parents:* High school Student, Zimbabwe.

These sentiments highlights the importance of maintaining young people's privacy and confidentiality in mental healthcare, ensuring that they feel safe and supported.

**Parents as a source of young people's mental health problems.** Another significant negative aspect of parental involvement is that parents can sometimes be the source of young people's mental health problems. For instance, when a young person is being abused at home, involving the same parents can inhibit openness. One high school student noted this concern:

*Let's say I am being abused by a parent, when they come at school, I can't tell you anything because they will be looking at me. But when we are alone, I can talk to you:* High school Student, Zimbabwe.

This sentiment highlights how the presence of an abusive parent can potentially silence young people, preventing them from disclosing key information that could help in alleviating their mental health concerns.

Another student shared the same sentiments, however adding another concern-fear of retaliation if parents discover that they are getting help elsewhere.

*Because some of them are the ones who illtreat us. If they knew that we are getting help somewhere, they might even hate us. That would become a disadvantage to us:* Senior High School Student, Ghana.

This comment highlights that unfortunately, some parents, who may be the source of the problem, might react negatively when they discover that their child is seeking help. This may expose young people to further rejection, criticism, or even abuse, exacerbating their mental health issues.

Overall, these excerpts suggest that, while parental involvement is key in SBMH interventions, it can also be detrimental when parents are the source of young people's mental health challenges. This dichotomy in perspectives highlights the need for a balanced approach that acknowledges both the importance of parental involvement and the potential challenges they may pose to their children.

## Discussion

Our study explored parental involvement in SBMH interventions for young people in Zimbabwe and Ghana. We identified several processes, including routine parent-teacher meetings, interface meetings between parents and school-based mental healthcare providers, and direct parental participation in the intervention sessions. Specifically, our study highlights the role of routine parent-teacher meetings as a key mechanism of parental involvement in SBMH interventions. This is consistent with previous research which highlight that PTA meetings serve as a vital bridge between home and school, through enhancing mutual trust, communication and collaboration between parents and teachers [20,28,35,36]. Other researchers argue that mutual trust is vital, as it helps teachers and parents connect, understand each other and work together to bring about positive changes in a child's mental health [20,37]. However, research indicates a striking disparity in parent meeting attendance by income [19,20,26,29,38]. In low-income settings, meeting attendance is often below expected due to logistical barriers such as time, distance to school, and lack of transportation [29,38–40]. This highlights the need to promote alternative ways of parental involvement. In line with this, some of our study participants suggested that, for minor mental health issues, parents could simply be informed about their child's mental health concerns and the school's planned intervention. This approach ensures that both the severity of the mental health concerns and the logistical challenges, such as the distance from parents' homes to school are considered. Importantly, our findings suggest a contextual difference between Ghana and Zimbabwe regarding the structure and utilisation of PTA meetings. While Ghanaian schools had well-established and widely attended PTA meetings, Zimbabwean schools had less formalised PTA structures, which could limit the effectiveness of this platform for parental engagement in SBMH initiatives. This highlights the need for alternative or supplementary strategies to involve parents effectively, particularly in settings where PTA participation is less pronounced.

Additionally, participants highlighted the critical role of interface meetings between parents and SBMH providers as a key mechanism for effective parental involvement. Previous studies have shown how interface meetings play a significant role in improving parents' mental health literacy. This includes recognising mental disorders, understanding available treatments, and developing skills to support their children with mental health issues among others [41–46]. Additionally, interface meetings demonstrate a commitment to individualised care and support, acknowledging the unique needs of young people experiencing mental health challenges [41,42]. However, in low-resource settings, parents generally have limited knowledge about the causes, symptoms and treatment of mental health issues, which can hinder their ability to recognise mental health problems in their children [47–49]. Therefore, the interface meetings suggested by study participants are particularly valuable, as they help bridge this knowledge gap.

Furthermore, direct parental participation in therapy sessions emerged as another key mechanism for parental involvement in the current study. This "therapeutic alliance", as described in the clinical literature [50,51], empowers parents

to reinforce therapy skills at home-an element that participants in the current study found essential for ongoing support, recovery and well-being. As previously indicated that young people cannot make decisions independently [52–54], therefore parents become important sources of information for early prognosis and diagnosis of mental disorders [55–58]. Moreover, involving parents in the sessions has been shown to improve young people's adherence to treatment and recovery. For instance, interventions where parents directly participated, like the 'Stress-Busters' [59], and the Positive Thoughts and Action [60,61] reported high rates of recovery and positive parent-child interactions. Similar studies that integrated parents, such as cognitive behavioural therapy (CBT) [62] and attachment-based family therapy (ABFT) [63,64], reported significant reductions in depression, hopelessness and suicidal ideation among young people.

However, while parental involvement is key for SBMH interventions, it can present some significant challenges. As our findings indicate, parents may unintentionally breach their children's privacy and confidentiality or may be the source of their children's mental health problems. Specifically, some young people in our study expressed hesitance to open-up in the presence of their parents due to intrusive nature of parents, fear of judgement, discovery of sensitive personal issues, or worry about being blamed for their mental health challenges. This often stems from lack of a comfortable relationship between the parent and the child, which can potentially lead to discomfort and distrust. Previous research emphasises the need for confidential care, underscoring how parental involvement in mental healthcare can breach young people's privacy and confidentiality [65,66]. Other studies have highlighted that young people may need to exercise their agency in mental healthcare decisions [67,68], and excessive parental involvement can be counterproductive [69]. This is reflected in certain therapy guidelines, which suggest that young people should be consulted on whether they want their parent(s) or caregiver(s) involved in their mental healthcare journey [70,71]. However, in African contexts, where the current study pulled from, parental involvement in all aspects of children's life is culturally emphasised [72], and implementing such guidelines may be problematic. The strong emphasis on parental authority and family unity may conflict with the idea of young people exercising their agency and autonomy in mental healthcare. This creates a delicate situation for service providers. Therefore, there is need to balance between respecting cultural values and promoting young people's agency and autonomy in mental healthcare settings.

Additionally, some young people viewed their parents as the source of their mental health concerns, making their involvement problematic. This finding is consistent with previous studies linking adverse family experiences such as abuse, violence in the family or divorce to poor mental health among young people [58,73]. This finding has significant implications for SBMH interventions, which may be ineffective if family issues remain unresolved [58]. Thus, not only young people (as clients of SBMH interventions), but also their parents, guardians or caregivers need support for these interventions to be effective.

Our study has some important limitations to note. Firstly, participants who were more engaged or have vested interest in mental health may be overrepresented in our sample. Secondly, the qualitative nature of our study does not allow for the identification of temporal or causal relationships between parental involvement and success of SBMH interventions. Thirdly, our study is context-specific, focusing only on Zimbabwe and Ghana, which may limit the generalizability of our findings to other settings. For instance, factors such as societal attitudes and norms toward mental health may vary across regions and may influence parental involvement differently. However, one of the major strengths of our study is conducting this study in two low-income settings which are predominantly underrepresented in theory and practice, thus offering insight for SBMH in these regions.

## Conclusions

Our study explored parental involvement in SBMH interventions for young people in Zimbabwe and Ghana. The findings suggest that routine parent-teacher meetings, interface meetings between parents and school-based mental healthcare providers, and direct parental participation in sessions may be a way to involve parents in SBMH interventions. However, the study also revealed some challenges of parental involvement, including the potential of parents to exacerbate

young people's mental health issues, particularly when privacy and confidentiality concerns arise, or when parents are the source of the mental health problems. Our findings have several implications for both practice and policy. First, school-based mental healthcare providers could prioritise regular and meaningful engagement with parents, treating them as partners in SBMH intervention development and implementation. This might involve increasing the frequency and quality of interface meetings, exploring alternative forms of parental involvement for those who face logistical barriers, and ensuring that interventions are flexible enough to accommodate situations where parental involvement may not be appropriate. Additionally, there is need for tailored approaches that consider different individual circumstances of young individuals, particularly those who may not benefit due to privacy concerns or adverse family dynamics. For future research, it is key to explore innovative mechanisms to parental involvement that are both effective and sensitive to the unique challenges faced by young people. This could involve exploring community-based or digital alternatives that do not require physical presence, as well as developing guidelines for parental involvement in ways that protects young people's privacy. It is also critically important for future studies to explore the impact of family dynamics that may contribute to mental health challenges. This could provide valuable insights into how to better support both young people and their families, thus enhancing the overall effectiveness of SBMH interventions.

## Supporting information

**S1 Checklist. PLOSOne_Human_Subjects_Research_Checklist 2511.**
(DOCX)

**S1 Questionnaire. Inclusivity-in-global-research-questionnaire.**
(DOCX)

## Acknowledgments

In Ghana, we acknowledge the support of the National School Health Education Programme Division, the Directors of health Services Navrongo, and the Mental Health Authority of Ghana. In Zimbabwe, we acknowledge the support of Ministry of Primary and Secondary Education, the Harare City Health Department, the Ministry of Health and Child Care, and the Friendship Bench.

## Author contributions

**Conceptualization:** Rufaro Hamish Mushonga, Rebecca Jopling, Benedict Weobong, Moses Kumwenda.

**Data curation:** Rufaro Hamish Mushonga, Rebecca Jopling, Franklin Glozah, Tiny Tinashe Kamvura, Suzanne Dodd, Denford Gudyanga, Arnold Maramba, Edith Dambayi, Christopher Abio Ayuure, Tarisai Bere, Fabian Sebastian Achana, Lucy Owusu, Moses Kumwenda.

**Formal analysis:** Rufaro Hamish Mushonga, Rebecca Jopling, Franklin Glozah, Tiny Tinashe Kamvura, Suzanne Dodd, Denford Gudyanga, Arnold Maramba, Edith Dambayi, Christopher Abio Ayuure, Tarisai Bere, Fabian Sebastian Achana, Lucy Owusu, Moses Kumwenda.

**Funding acquisition:** Dixon Chibanda, Melanie Abas.

**Methodology:** Rufaro Hamish Mushonga, Rebecca Jopling, Franklin Glozah, Tiny Tinashe Kamvura, Suzanne Dodd, Denford Gudyanga, Arnold Maramba, Edith Dambayi, Christopher Abio Ayuure, Tarisai Bere, Fabian Sebastian Achana, Lucy Owusu, Dixon Chibanda, Melanie Abas, Benedict Weobong, Moses Kumwenda.

**Project administration:** Suzanne Dodd, Melanie Abas.

**Resources:** Dixon Chibanda, Melanie Abas.

**Software:** Denford Gudyanga.

**Supervision:** Dixon Chibanda, Melanie Abas, Benedict Weobong, Moses Kumwenda.

**Writing – original draft:** Rufaro Hamish Mushonga, Rebecca Jopling, Franklin Glozah, Tiny Tinashe Kamvura, Suzanne Dodd, Denford Gudyanga, Arnold Maramba, Edith Dambayi, Christopher Abio Ayuure, Tarisai Bere, Fabian Sebastian Achana, Lucy Owusu, Benedict Weobong, Moses Kumwenda.

**Writing – review & editing:** Rufaro Hamish Mushonga, Rebecca Jopling, Franklin Glozah, Tiny Tinashe Kamvura, Suzanne Dodd, Denford Gudyanga, Arnold Maramba, Edith Dambayi, Christopher Abio Ayuure, Tarisai Bere, Fabian Sebastian Achana, Lucy Owusu, Benedict Weobong, Moses Kumwenda.

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
