## [Decision Letter · Decision Letter 0]

11 Mar 2025

PONE-D-24-48920Parental Involvement in School-Based Mental Health interventions for Young People in low resource settings: A Qualitative study from Zimbabwe and GhanaPLOS ONE

Dear Dr. Mushonga,

Thank you for submitting your manuscript to PLOS ONE. After careful consideration, we feel that it has merit but does not fully meet PLOS ONE’s publication criteria as it currently stands. Therefore, we invite you to submit a revised version of the manuscript that addresses the points raised during the review process.

We look forward to receiving your revised manuscript.

Kind regards,

Engelbert A. Nonterah, MD, PhD

Academic Editor

PLOS ONE

Journal Requirements:

2. Please include a complete copy of PLOS’ questionnaire on inclusivity in global research in your revised manuscript. Our policy for research in this area aims to improve transparency in the reporting of research performed outside of researchers’ own country or community. The policy applies to researchers who have travelled to a different country to conduct research, research with Indigenous populations or their lands, and research on cultural artefacts. The questionnaire can also be requested at the journal’s discretion for any other submissions, even if these conditions are not met.  Please find more information on the policy and a link to download a blank copy of the questionnaire here: https://journals.plos.org/plosone/s/best-practices-in-research-reporting. Please upload a completed version of your questionnaire as Supporting Information when you resubmit your manuscript.”

4. In the online submission form you indicate that your data is not available for proprietary reasons and have provided a contact point for accessing this data. Please note that your current contact point is a co-author on this manuscript. According to our Data Policy, the contact point must not be an author on the manuscript and must be an institutional contact, ideally not an individual. Please revise your data statement to a non-author institutional point of contact, such as a data access or ethics committee, and send this to us via return email. Please also include contact information for the third party organization, and please include the full citation of where the data can be found.

Reviewers' comments:

Reviewer's Responses to Questions

**Comments to the Author**

1. Is the manuscript technically sound, and do the data support the conclusions?

Reviewer #1: Yes

Reviewer #2: Yes

2. Has the statistical analysis been performed appropriately and rigorously? 

Reviewer #1: N/A

Reviewer #2: N/A

3. Have the authors made all data underlying the findings in their manuscript fully available?

Reviewer #1: Yes

Reviewer #2: Yes

4. Is the manuscript presented in an intelligible fashion and written in standard English?

Reviewer #1: Yes

Reviewer #2: Yes

5. Review Comments to the Author

Reviewer #1: Parental Involvement in School-Based Mental Health interventions for Young People in

low resource settings: A Qualitative study from Zimbabwe and Ghana

This qualitative study explored the mechanisms for effectively involving parents in SBMH interventions for young people in Zimbabwe and Ghana (low resource setting).

Overall

• The manuscript is well written, very clear and adds knowledge on parental involvement in school-based mental health interventions for young people in a resource limited setting. The title is appropriate and the aim is clearly stated. The research is important to the field. The structure and language are appropriate.

Abstract

• It is very clear and it highlights the key findings of the study.

Introduction

• The introduction is sufficient, clear and well organized.

• The subject matter is well introduced and laid a solid foundation for the manuscript.

• I think it could be helpful to add a sentence highlighting the fact that parental involvement is currently not part of the school based mental health interventions in Zimbabwe and Ghana.

Methodology

• The methods are reproducible and well described.

• Correct terms were used to describe the study population

• Clear inclusion and exclusion criteria.

• It would be helpful to explain why the different age groups for the two countries (5-24 in Zimbabwe and 15-18 in Ghana).

• Please shed some light on why CMDs are common among young people in Navrongo, Ghana.

• For table 1 consider using sex instead of gender, gender is a social construct that refers to the behaviors and role and identities of people.

• For table 1 is there a difference between the zeros and the dashes?

Results and discussion

• The results are well presented.

• There is general flow and logic on justification of interpretation of results and the conclusion.

• Clear strengths and limitations.

Conclusion

• Precise and to the point.

References, tables and figures

• The references are numbered correctly and citations are appropriate.

• For the references please check for inconsistencies i.e. some journal names are written in full and others are shortened. References number 5 and 10 (W.H.O and World Health Organization) etc.

• The table is well labeled.

• There is presentation consistency.

Reviewer #2: This is a very well-written and organized manuscript that addresses a gap in the literature regarding an important issue of adolescent mental health. The methodology is sound and described in depth. There are only a few minor suggestions that I would make to the authors:

- In Methods (Inclusion and Exclusion Criteria), authors mention "active major mental disorder" as an exclusion criterion. Please provide additional details on how that was defined - whether by RA designation, previous diagnosis, use of pharmacotherapy, etc.

- In Methods, consider providing additional details that can help contextualize the socioeconomic status of the schools/student participants. For example, if public vs private school. This can be useful since the Discussion mentions logistical barriers to parental involvement.

- In Results/Discussion, consider adding some comparison of responses between the two study sites in Zimbabwe and Ghana. While the responses from both sites seem to reinforce each other, it would be interesting to mention if there were any differences noted between the sites.

- In Results, the level of subheadings and their formatting needs to be fixed.

6. PLOS authors have the option to publish the peer review history of their article (what does this mean? ). If published, this will include your full peer review and any attached files.

**Do you want your identity to be public for this peer review?** For information about this choice, including consent withdrawal, please see our Privacy Policy .

Reviewer #1: No

Reviewer #2: **Yes: ** Masih A Babagoli

---

## [Author Response · Author response to Decision Letter 1]

28 Mar 2025

Response to Reviewers

Editor Comments

Response: The manuscript has now been formatted as per PLOS ONE’s style requirements

2. Please include a complete copy of PLOS’ questionnaire on inclusivity in global research in your revised manuscript. Our policy for research in this area aims to improve transparency in the reporting of research performed outside of researchers’ own country or community. The policy applies to researchers who have travelled to a different country to conduct research, research with Indigenous populations or their lands, and research on cultural artefacts. The questionnaire can also be requested at the journal’s discretion for any other submissions, even if these conditions are not met. Please find more information on the policy and a link to download a blank copy of the questionnaire here: https://journals.plos.org/plosone/s/best-practices-in-research-reporting. Please upload a completed version of your questionnaire as Supporting Information when you resubmit your manuscript.”

Response: A complete copy of PLOS’ questionnaire on inclusivity in global research has been included as supporting information

Response: All additional details requested regarding consent/assent has been provided in the Methods section

4. In the online submission form you indicate that your data is not available for proprietary reasons and have provided a contact point for accessing this data. Please note that your current contact point is a co-author on this manuscript. According to our Data Policy, the contact point must not be an author on the manuscript and must be an institutional contact, ideally not an individual. Please revise your data statement to a non-author institutional point of contact, such as a data access or ethics committee, and send this to us via return email. Please also include contact information for the third party organization, and please include the full citation of where the data can be found.

Response: The data underlying the results presented in this study are available from the Institute of Psychiatry, Psychology & Neuroscience, King's College London, United Kingdom (16 De Crespigny Park, London SE5 8AF). The point of contact for the data is Melanie Abas, the Principal Investigator of the project at KCL. As per our ethical approvals, access to the data is restricted to the study team, who are co-authors of this paper. Since participant consent was not obtained for data sharing beyond the study team, we are unable to deposit the data in a public repository or share it with external individuals.

Response: Reference list has been reviewed

Reviewer 1 Comments

I think it could be helpful to add a sentence highlighting the fact that parental involvement is currently not part of the school based mental health interventions in Zimbabwe and Ghana.

Response: The statement requested was added in the introduction section of the manuscript

It would be helpful to explain why the different age groups for the two countries (15-24 in Zimbabwe and 15-18 in Ghana).

Response: Additional information on why the different age groups for the two countries (15-24 in Zimbabwe and 15-18 in Ghana) was provided.

Please shed some light on why CMDs are common among young people in Navrongo, Ghana.

Response: The requested information was provided

For table 1 consider using sex instead of gender, gender is a social construct that refers to the behaviors and role and identities of people.

Response: This was resolved

For table 1 is there a difference between the zeros and the dashes?

Response: This was resolved

For the references please check for inconsistencies i.e. some journal names are written in full and others are shortened. References number 5 and 10 (W.H.O and World Health Organization) etc.

Response: This was resolved

Reviewer 2 Comments

In Methods (Inclusion and Exclusion Criteria), authors mention "active major mental disorder" as an exclusion criterion. Please provide additional details on how that was defined - whether by RA designation, previous diagnosis, use of pharmacotherapy, etc.

Response: Additional details requested were provided

In Methods, consider providing additional details that can help contextualize the socioeconomic status of the schools/student participants. For example, if public vs private school. This can be useful since the Discussion mentions logistical barriers to parental involvement.

Response: Additional details with regards to socio-economic status of the schools and participants were provided.

In Results/Discussion, consider adding some comparison of responses between the two study sites in Zimbabwe and Ghana. While the responses from both research sites seem to reinforce each other, it would be interesting to mention if there were any differences noted between the sites.

Response: Slight comparisons were provided in the results section (PTA meetings) and discussion section as largely the results from both sites reinforce each other.

In Results, the level of subheadings and their formatting needs to be fixed.

Response: Resolved

---

## [Editor Report · Decision Letter 1]

1 Apr 2025

Parental Involvement in School-Based Mental Health interventions for Young People in low resource settings: A Qualitative study from Zimbabwe and Ghana

PONE-D-24-48920R1

Dear Dr. Rufaro Hamish Mushonga,

We’re pleased to inform you that your manuscript has been judged scientifically suitable for publication and will be formally accepted for publication once it meets all outstanding technical requirements.

Kind regards,

Engelbert A. Nonterah, MD, PhD

Academic Editor

PLOS ONE
---

## [Editor Report · Acceptance letter]

PONE-D-24-48920R1

PLOS ONE

Dear Dr. Mushonga,

I'm pleased to inform you that your manuscript has been deemed suitable for publication in PLOS ONE. Congratulations! Your manuscript is now being handed over to our production team.

Kind regards,

on behalf of

Dr. Engelbert Adamwaba Nonterah

Academic Editor

PLOS ONE